# Liquid Biopsies in Colorectal Liver Metastases: Towards the Era of Precision Oncologic Surgery

**DOI:** 10.3390/cancers14174237

**Published:** 2022-08-31

**Authors:** Diamantis I. Tsilimigras, Ioannis Ntanasis-Stathopoulos, Timothy M. Pawlik

**Affiliations:** 1Department of Surgery, Division of Surgical Oncology, The Ohio State University Wexner Medical Center and James Comprehensive Cancer Center, Columbus, OH 43210, USA; 2Department of Clinical Therapeutics, School of Medicine, Alexandra General Hospital, National and Kapodistrian University of Athens, 11527 Athens, Greece

**Keywords:** ctDNA, CTCs, colorectal cancer, liver metastases, liquid biopsy, RAS

## Abstract

**Simple Summary:**

Liquid biopsies are increasingly gaining attention in the field of cancer therapeutics and, in particular, in the field of oncologic surgery. Liquid biopsy testing could help improve prognostication, detect recurrences early and monitor tumor evolution in the context of cancer therapies. This review aimed to provide an overview of liquid biopsies and focused on their utility among patients with CRLM.

**Abstract:**

Tumor mutational analysis has been incorporated into the management of patients with CRLM since it can provide valuable prognostic information as well as guide peri-operative systemic treatment. Unlike tumor biopsy, liquid biopsy has emerged as a promising, non-invasive alternative that can detect cell-derived markers from a variety of body fluids and might better characterize all subclones present at a specific time point and allow sequential monitoring of disease evolution. Although not currently considered standard of care, an increasing number of cancer centers are nowadays routinely using liquid biopsies in the treatment of CRLM patients with promising results. The current review provides an overview of liquid biopsies in cancer therapeutics and focuses on the application of this relatively new approach on patients with CRLM.

## 1. Introduction

Colorectal cancer (CRC) is the third most common cancer worldwide. Despite advances in diagnosis and treatment of CRC, approximately one-half of CRC patients will develop colorectal liver metastases (CRLM) during the course of their disease [1,2,3]. Surgery remains the best curative-intent treatment option for patients with resectable CRLM, yet the incidence of recurrence remains high despite advances in systemic chemotherapy over the last several decades [4]. Strategies to mitigate recurrence include peri-operative chemotherapy, targeted biologic agents, as well as tumor-derived biomarkers that enhance prognostication and the ability to predict response to therapy.

Tumor mutational analysis has been incorporated into current clinical practice since it can provide valuable prognostic information, as well as guide peri-operative systemic treatment among patients with resectable CRLM. The current literature suggests that approximately 40–45% of CRLM patients harbor a somatic RAS mutation, whereas the prevalence of BRAF mutation is 5–12% [5]. RAS mutations have been associated with worse overall and recurrence-free survival, as well as resistance to anti-epidermal growth factor receptor (EGFR) therapy. Although tumor tissue DNA mutation analysis is the gold standard, tumor genotyping requires either a resected specimen or a conventional tissue biopsy. Nevertheless, tissue biopsy is not always performed in clinical practice, and does not always accurately represent the genetic heterogeneity of the tumor. Rather, it provides a single static snapshot of a small fragment of tumor.

Unlike tumor biopsy, liquid biopsy has emerged as a promising, non-invasive alternative that can detect cell-derived markers from a variety of body fluids, including blood, saliva, urine, and cerebrospinal fluid. The analysis of cell-derived biomarkers in body fluids is known as liquid biopsy [6]. Cell-derived markers include circulating tumor cells (CTCs), extracellular nucleic acids (i.e., cell-free DNA [cfDNA], mRNA, micro-RNA, long non-coding RNAs, circular RNAs, small non-coding RNAs or PIWI-interacting RNAs [piRNAs]), exosomes as well as glycoproteins and antigens (i.e., CEA, CA19-9, CA-125 etc.). Oral and gut microbiome-associated serum metabolites can also be assessed with liquid biopsies along with tumor-educated platelets (TEPs) [7,8]. cfDNA is a fragment of DNA released into the plasma following apoptosis of normal or tumor cells. Circulating tumor DNA (ctDNA) is a subset of cfDNA secreted into the bloodstream by primary tumors, metastases or even CTCs. Both cfDNA and ctDNA carry genome-wide DNA information and can effectively overcome the issues of the spatial heterogeneity of tumors encountered with the traditional tissue biopsy [9]. miRNAs are small, single-stranded nucleotides (approximately 20–22 nucleotides) that might act as oncogenes or tumor suppressors through various mechanisms [8]. Circular RNAs represent non-coding RNAs with covalently closed loop that can regulate transcription and alternative splicing by interacting with RNA-binding proteins. LncRNAs are non-coding RNAs, usually longer than 200 nucleotides, that do not encode proteins but are able to up- or down-regulate certain oncogenic or tumor suppressor genes. In contrast, piRNAs are small non-coding nucleotides (usually 24–32 nucleotides in length) that have been associated with several gene regulation mechanisms including transposon silencing, epigenetic programming, DNA rearrangements, mRNA turnover, and translational control [8]. Exosomes are heterogeneous extracellular vesicles enclosed by a cholesterol-rich lipid bilayer that carry a variety of biologically active molecules that mirror the composition of their originating cells including nucleic acids (DNA, RNA, miRNAs, lncRNA etc), proteins and lipids [7]. Metagenomic analysis of oral and gut microbiota also has the ability to discriminate cancer versus healthy individuals, given that certain groups of oral pathogens seem to be more abundant in patients with CRC (Fusobacterium, Porphyromonas, and Treponema) and have been postulated to be involved in CRC tumorigenesis [7]. Finally, TEPs are an emerging concept where cancer cells activate platelets which, in turn, promote cancer progression through upregulation of certain factors, including VEGF, PDGF, and PF4 that remain elevated in CRC patients [10,11].

Liquid biopsy has the potential to detect minimal residual disease (MRD) [12] that is not evident in surveillance imaging, as well as direct targeted therapies based on tumor genotyping and monitor recurrence in real time. In turn, liquid biopsy has been increasingly used in clinical practice and can provide a more comprehensive molecular profile of cancer compared to a single tissue biopsy. The recently published results of the DYNAMIC clinical trial have demonstrated that ctDNA-based treatment decisions regarding the administration of adjuvant chemotherapy in patients with stage II colon cancer do not compromise patient outcomes. Recurrence-free survival at 3 years was 86% for patients with positive ctDNA postoperatively who received adjuvant treatment, whereas it reached 93% for individuals with negative ctDNA after surgery who did not receive chemotherapy [13]. The current review provides an overview of liquid biopsies in cancer therapeutics mainly focusing on the detection of CTCs and ctDNA among patients with CRLM.

## 2. The Biology behind Liquid Biopsies: CTCs and ctDNA

### 2.1. CTCs

Tumor cells are released from primary or metastatic tumors into the bloodstream [14]. It is suggested that the half-life of CTC in the bloodstream is only 1–2.4 h [15]. The exact mechanism behind the release of CTC into the bloodstream is currently unknown. Apoptotic or fragmented tumor cells have previously been described in the peripheral blood of cancer patients suggesting an unfavorable environment in which tumor cells are released [14]. Surviving CTCs are cleared through extravasation into secondary organs. For example, previous analyses of mesenteric and peripheral veins among patients with CRC demonstrated that the liver captures the majority of tumor cells released by CRC [16], which is in line with preclinical data in the field [17]. CTCs have also been involved in disease progression by forming aggregates with activated platelets and macrophages [18], which facilitates attachment to the endothelium and, in turn, the development of distant metastases [19]. Several chemokines, including CXCR4, CCR4, CCR7, and CCR9, also play a significant role in the migration of metastatic cells into the circulation [20]. The tumor microenvironment (TME) is crucial in tumor progression and analysis of TME though liquid biopsies can provide significant information on prognosis and response to treatment [21]. Of note, the detection of CTCs among patients who have undergone primary tumor resection might indicate that tumor cells recirculate from secondary metastatic sites into the bloodstream and is currently considered a poor prognostic indicator [15,22]. A high number of baseline CTCs in patients with metastatic CRC has also been associated with adverse patient and disease characteristics, including worse performance status, stage IV at diagnosis, at least three metastatic sites, and elevated CEA levels [23].

CTC assays usually start with an enrichment step that increases the concentration of CTCs by several log units and enables easier detection of single tumor cells [24]. CTCs can be positively selected through the use of epithelial markers that are not expressed on the surrounding mesenchymal blood cells (e.g., EpCAM, mucin-1, HER2 or EGFR) (i.e., positive enrichment) or negatively selected by depletion of normal circulating cells using markers that are lacking from tumor cells (e.g., CD45 for leukocytes) (i.e., negative enrichment). Besides biologic properties (i.e., expression of protein markers), enrichment might also occur on the basis of different physical properties of tumor versus normal mesenchymal cells (i.e., size, density, deformability, or electric charges). Two types of CTC enrichment, positive or negative, can be achieved taking into account a combination of physical and biologic properties in the same device. Following enrichment, CTCs can be identified using immuno-cytological molecular or functional approaches to further identify and characterize CTCs [25]. Although researchers have utilized CTC cultures/cells line and xenografts in the past [26,27], this method has significant disadvantages including the need for a significant amount of time and hundreds of CTCs to establish a cell line or xenograft, thus limiting this approach to only a subset of patients with advanced disease (Figure 1A). A different non-enrichment methodology has also been described in the treatment of metastatic prostate cancer. In brief, immunofluorescent staining can be used to identify CTCs that are either positive or negative for androgen-receptor splice variant 7 (AR-V7) and is now commercially available as AR-V7 and used in clinical practice to guide treatment among individuals with metastatic castration-resistant prostate cancer (mCRPC). In turn, patients with mCRPC with CTC nuclear expression of AR-V7 protein are now recommended taxane therapy over androgen receptor signaling (ARS) inhibitors [28,29].

### 2.2. ctDNA

Tumor DNA can be released from primary tumors, CTCs, or micro- or macro-metastases into the blood of cancer patients. ctDNA is mainly derived from apoptotic or necrotic tumor cells that release fragmented DNA into the circulation [24]. Apart from ctDNA, circulation contains cfDNA normally released by non-malignant cells that have completed their life cycle. This normal cfDNA can dilute the ctDNA among patients with cancer, especially individuals who have undergone tissue-damaging therapies such as surgery, chemotherapy, or radiotherapy [24]. Although the mechanism behind the clearance of cfDNA is not fully understood, cfDNA is considered to have a short half-life of approximately 16 min [31] and is cleared by the liver and kidneys [32]. In turn, cfDNA clearance might be impaired in cancer patients with renal dysfunction, which may inflate the ctDNA levels in these patients [33]. ctDNA has also been postulated to affect the biology of host cells by incorporating into their genome (i.e., geno-metastasis) [33]. For example, Trejo-Becerril et al. demonstrated the ability of cfDNA to induce in vitro cell transformation and tumorigenesis by treating NIH3T3 recipient murine cells with the serum of CRC patients and supernatant of SW480 human cancer cells [34]. Interestingly, cell transformation and tumorigenesis were not evident when serum and supernatants were depleted of DNA. These data support the argument that cancer cells secrete into the circulation biologically active DNA that has oncogenic properties and might contribute to tumor progression. As such, ctDNA may represent a promising target to develop novel antitumor therapies.

Different methods have been developed to detect ctDNA, including BEAMing Safe-SeqS, TamSeq, and droplet-based digital PCR (ddPCR), either by detecting single-nucleotide mutations in ctDNA or by whole-genome sequencing/next-generation sequencing (NGS) that can establish copy-number changes [35,36]. In general, existing technologies can be divided into targeted approaches that identify mutations in a set of predefined genes (e.g., KRAS in the context of EGFR blockade by antibodies) or untargeted approaches (e.g., array-CGH, whole-genome sequencing, or exome sequencing) that screen the genome and discover new genomic aberrations, including those that confer resistance to a specific targeted therapy (Figure 1B) [37]. In general, targeted approaches are associated with a higher sensitivity to detect ctDNA compared with untargeted approaches, yet are limited by the number of predetermined mutations able to be examined with each test [38]. It is estimated that less than 5% of cfDNA consists of ctDNA released by tumor cells [39]. In turn, investigators have been recently focusing on developing ultrasensitive technologies that will be able to identify accurately even the smallest amounts of ctDNA within normal cfDNA, a critical step for early detection of cancer or MRD [40].

## 3. Liquid Biopsies in CRLM: Experience and Utility

Recurrence among patients with CRLM still remains high (approximately 50% within 2 years) even after curative-intent resection. Standard of care surveillance includes periodic CT scans and measurement of tumor-specific markers, such as CEA. Indeterminate CT findings remain a significant challenge for oncologists and can often lead to delayed interventions [41]. In addition, serum markers have been far from perfect in identifying recurrences in the postoperative period. Several clinical risk scores have been developed, including models that take molecular markers (i.e., RAS and BRAF mutations) into account, to aid in the stratification of patients relative to risk of recurrence, yet their performance has been suboptimal in different cohorts. Currently, there are no validated biomarkers to tailor surveillance strategies according to the individual risk of recurrence, as well as help guide the use of chemotherapy among CRLM patients. Patients with CRLM usually receive therapy based on the molecular traits of the primary tumor. Nevertheless, mutations occur constantly during disease progression and the characteristics of metastatic tumors might change as part of their molecular evolution. Molecular analysis of primary tumors does not always suffice to stratify individuals accurately and guide physicians to the most promising therapy. Re-analysis of metastatic lesions, although possible, is rather limited by the inter-lesion heterogeneity that does not always capture the entire spectrum of mutational changes. By analyzing tumor cells or ctDNA in blood samples, liquid biopsies might better characterize all subclones present at a specific time point and allow sequential monitoring of disease evolution [42]. In turn, liquid biopsies have emerged as a promising tool among oncologists specializing in different types of tumors including hepato-pancreato-bilary cancers [43,44,45]. Although not currently considered standard of care, an increasing number of cancer centers are routinely using liquid biopsies in the treatment of CRLM patients with promising results.

### 3.1. Prediction of Recurrence following CRLM Resection

The MD Anderson group recently published their experience with liquid biopsy results among individuals who had hepatectomy for CRLM [46]. Of note, the authors analyzed plasma from 63 patients drawn postoperatively using next generation sequencing analysis to detect somatic mutations in 70 genes. Patients with a positive liquid biopsy (i.e., at least one gene mutation) postoperatively had significantly worse 2-year OS compared with individuals with negative liquid biopsy (70% vs. 100%; *p* = 0.005) [46]. Importantly, a higher number of gene mutations detected in liquid biopsy correlated with worse OS (four or more gene mutations; 2-year OS: 41%), whereas a positive liquid biopsy-rather than elevated serum CEA-appeared to be strongly associated with CT evidence of metastatic disease [46]. In a more recent study, patients with ctDNA+ were twice as likely to have a RAS + TP53 co-mutation–considered an important prognostic factor-versus individuals with ctDNA- following CRLM resection (47% vs. 23%, HR = 2.04, *p* = 0.01) [47]. Of note, recurrence rates within 1 year of hepatectomy were 94% among ctDNA+ individuals versus 49% among ctDNA- patients (*p* = 0.003) [47]. Perhaps more interesting, ctDNA+ was the only independent risk factor for early recurrence (i.e., within 6 months) following CRLM recurrence (HR 11.8, 95%CI 2.3–59.8) [47].

In another prospective observational study, Reinert et al. analyzed ctDNA for mutations in the genes APC, BRAF, KRAS, NRAS, PIK3CA and TP53 among 115 patients undergoing resection for CRLM. The authors collected blood samples prior to surgery, postoperatively at day 30 (POD30) and every 3 months up to 3 years [48]. Among patients with positive ctDNA on POD30, recurrence was as high as 100% versus 55.6% among patients who were ctDNA negative (HR = 7.6, 95%CI 3.0–19.7, *p* < 0.001). A positive ctDNA at 3 months after CRLM resection was a strong predictor for subsequent disease relapse [49]. On multivariable analysis, ctDNA status, and not primary tumor N stage, or CEA, was the only significant prognostic factor associated with recurrence (*p* < 0.001). Perhaps more interestingly, inconclusive CT scans during surveillance were observed in 21 out of 68 patients (30.1%), which led to significant delays in intervention (patients with indeterminate findings 3.8 months vs. no indeterminate findings 10 months, *p* < 0.001). Among individuals with indeterminate imaging during surveillance, positive ctDNA status was 100% predictive of recurrence (PPV 100%). Of note, among individuals who relapsed, ctDNA relapse almost always preceded clinical relapse (median time to ctDNA relapse: 3.1 months vs. median time to clinical relapse: 6.1 months), highlighting the importance of serial assessment of ctDNA to detect MRD during follow-up, especially in the setting of indeterminate surveillance imaging findings [48,49].

Another prospective cohort study from Denmark evaluated the predictive value of postoperative ctDNA in 96 patients who underwent resection of CRLM with curative intent [50]. The assessment was performed by means of the methylation-based ddPCR TriMeth. The detection of ctDNA at any time following surgery was associated with significantly shortened RFS, whereas its predictive value surpassed other standard predictors for recurrence including CEA and clinical variables. Importantly, ctDNA emerged as an early marker of disease progression preceding imaging evidence of recurrence in almost half of patients with ctDNA-positive findings. CtDNA assessment also had high positive and negative predictive values in the setting of inconclusive imaging results, which may help in the decision-making for subsequent surveillance. Longitudinal analysis of plasma samples revealed that the dynamics of ctDNA may also predict OS [50]. Furthermore, among 23 patients enrolled in a phase 3 prospective clinical trial who underwent resection of CRLM with RAS mutations following neoadjuvant systemic treatment, individuals with detectable ctDNA by ddPCR postoperatively had a median RFS of 4.8 months versus 12.1 months among patients with undetectable ctDNA. Postoperative ctDNA status was also associated with pathologic response [51].

In a single center retrospective study from Japan, Kobayashi et al. recently investigated the impact of preoperative ctDNA on survival outcomes among patients who underwent hepatectomy for solitary resectable CRLM [52]. Among 40 patients with pre-hepatectomy plasma analysis, 32 (80%) had at least one somatic ctDNA alteration, whereas the remaining 8 (20%) patients had undetectable ctDNA [52]. RFS was significantly shorter among patients with positive ctDNA (HR = 7.6, *p* = 0.02). Of note, among patients with preoperatively undetectable ctDNA, only one patient experienced a recurrence after a median follow up of 39 months [52]. Of note, a different retrospective study from the Czech Republic did not find any association between the presence of pre-operative KRAS mutations by ddPCR in the ctDNA and RFS following CRLM resection. Interestingly, a multivariate analysis noted that high levels of preoperative KRAS fractional abundance and high CEA levels predicted for inferior OS [53]. These results were also confirmed by a meta-analysis of 12 studies which demonstrated that OS (HR 2.47, 95%CI 1.74–3.51) and PFS (HR 2.07, 95%CI 1.44–2.98) were worse among patients with CRLM and detectable CTC when compared with CTC-negative individuals [54].

Regarding CTCs, a prospective study from Canada suggested that a CTC number above 3 was associated with shortened PFS and OS among 63 patients who underwent resection for CRLM. Of note, only CTCs detected in the hepatic vein had a prognostic value, compared with CTCs detected in the peripheral bloodstream intraoperatively [55].

### 3.2. Prediction of Resistance to Treatment

In patients with CRLM, systemic treatment with monoclonal antibodies directed against EGFR is one of the therapeutic options considered for CRLM patients with wild-type RAS and BRAF status. Analysis of KRAS/BRAF status has, therefore, been routinely implemented in clinical practice prior to anti-EGFR therapy. Although tissue-based DNA mutation analysis is currently the gold standard approach, it might not accurately represent the genetic heterogeneity of the tumor. In a multicenter, prospective clinical trial, van’t Erve et al. investigated the incorporation of liquid biopsy ctDNA analysis to traditional tissue DNA analysis among 100 patients with liver-only unresectable CRLM [56]. There was an excellent concordance rate of 93% between tissue DNA and liquid biopsy ctDNA mutation status. In addition, by utilizing a decision tree analysis, a consecutive RAS/BRAF ctDNA analysis followed by tissue DNA analysis in the setting of liquid biopsy-negative resulted in an increase in the proportion of patients with RAS/BRAF alterations and, therefore, the accuracy of determining patient eligibility for anti-EGFR therapy. Of note, the combination of liquid biopsy and tissue biopsy in case of negative liquid biopsy appeared to be less expensive than tissue-based analysis alone ($539 vs. $683 per patient) [56]. The authors concluded that liquid biopsy testing for RAS/BRAF mutations is a cost-saving complementary approach to routine tissue-based DNA mutation analyses and should be incorporated in clinical practice [56]. In another study that included 76 patients with metastatic colorectal cancer, 32 of whom had liver metastases, the detection of RAS/BRAF mutations with next-generation sequencing (NGS) in the ctDNA was associated with short PFS during the first-line chemotherapy [57].

### 3.3. Real-Time Monitoring of Response to Therapy

Serial ctDNA mutation analyses can be used to monitor the effectiveness of therapy in real time, identify new mutations and resistance mechanisms and, therefore, guide subsequent treatment. In a prospective study, Wang et al. investigated the dynamic changes in peri-operative ctDNA among 91 patients undergoing resection for CRLM [58]. The authors demonstrated that decreasing pre-operative ctDNA levels during pre-operative chemotherapy predicted better tumor response rate, suggesting a potential role of dynamic ctDNA monitoring in tailoring the intensity of pre-operative treatment [58]. In addition, analysis of ctDNA during adjuvant chemotherapy following CRLM resection demonstrated that decreased ctDNA variant allele frequency (VAF) was associated with lower rates of recurrence when compared with increased ctDNA VAF (63.6% vs. 92.3%), suggesting that serial analysis of ctDNA in CRLM after hepatectomy could potentially be used as a real-time marker to determine the subgroups of patients who would or would not benefit from adjuvant chemotherapy [58].

In another prospective, multicenter cohort study, Tie et al. analyzed 54 patients with resectable CRLM who either received upfront resection (*n* = 31) or neo-adjuvant chemotherapy followed by resection (*n* = 23) [59]. The authors analyzed plasma samples for serial ctDNA mutations in 15 genes (SMAD4, TP53, AKT1, APC, BRAF, CTNNB1, ERBB3, FBXW7, HRAS, KRAS, NRAS, PIK3CA, PPP2R1A, RNF43, POLE) prior to and after surgery, during pre- and postoperative chemotherapy as well as during the follow-up period [59]. Patients with detectable postoperative ctDNA had worse RFS (HR = 6.3, 95%CI 2.58–15.2, *p* < 0.001) than individuals with undetectable ctDNA. Among individuals with persistently positive ctDNA who had serial ctDNA sampling during adjuvant chemotherapy (*n* = 11), three patients had ctDNA clearance of whom two remained disease-free at last follow up, whereas all remaining patients with persistently detectable ctDNA after adjuvant chemotherapy (*n* = 8) experienced recurrence [59]. The authors concluded that serial ctDNA analysis during adjuvant chemotherapy could be an early marker of treatment efficacy and could help guide postoperative treatment.

Furthermore, liquid biopsies may be used for monitoring response to neoadjuvant treatment and predict outcomes following CRLM resection. A prospective study evaluated both CTCs and KRAS ctDNA by ddPCR in 153 patients with CRLM eligible for surgical resection who received preoperative treatment [60]. Patients responding to neoadjuvant therapy had a gradual decrease in CTC counts and ctDNA detection level. The patients with detectable ctDNA following the completion of neoadjuvant treatment before surgery were less likely to undergo R0/R1 resection. Interestingly, the subset of patients with positive ctDNA before CRLM resection experienced worse OS compared with patients with negative ctDNA preoperatively [60].

Another prospective study evaluated the predictive role of methylation markers of 47 genes (AIT 47 gene panel) during the neoadjuvant treatment of 34 patients with CRLM prior to intended resection [61]. Peripheral blood plasma was collected at baseline and prior to each cycle of neoadjuvant chemotherapy. Of note, 26 of the 47 examined genes were elevated in all patients at baseline, whereas the traditional CEA and CA19—9 markers were elevated in 85% and 50% of patients, accordingly. Of note, although CEA levels correlated with tumor volume, SEPT9 and 24 markers of the AIT 47 gene panel displayed a stronger correlation with overall tumor volume than CEA [61]. The authors selected four methylation markers (SEPT9, BOLL, DCC and SFRP2) to be used for serial disease monitoring based on the following three criteria; 1. markers had to be positive in all patients at baseline, 2. markers should have a correlation coefficient with tumor volume that was superior to that of CEA and 3. selected markers should reflect a 100-fold change in tumor volume by a ΔCt (cycle of threshold) of ≥10 to allow for better discrimination of small differences in tumor volume [61]. Of note, patients that were finally operated upon had lower baseline methylation marker levels than patients who did not receive surgery reflecting a higher tumor burden in the latter group. Of note, the combination of all four markers showed a high accuracy in predicting operability (AUC > 0.88). Following administration of two rounds of chemotherapy, CEA and CA19-9 remained largely unchanged whereas methylation markers decreased in all patients, with more pronounced decrease noted among individuals who finally underwent surgery [61]. The sensitivities and specificities in predicting operability with methylation markers ranged from 82–91% and 95–100%, respectively. In addition, individuals with histopathologic response following chemotherapy tended to have decreased levels of methylation markers. The authors concluded that serial assessment of the methylation markers SEPT9, BOLL, DCC and SFRP2 predicted operability as early as at the beginning of the second cycle of therapy, as well as histopathological response [61].

### 3.4. Concordance Rates between Liquid and Tissue Biopsies

A fundamental issue in determining the clinical utility of liquid biopsies in the management of CRLM is establishing the concordance rate with tissue biopsies from primary and metastatic sites. The RAS mutational status of each patient with CRLM is key information in order to tailor treatment regimens. Liquid biopsies have been implemented in this setting in order to reduce turnaround times and initiate targeted therapies promptly.

A prospective multicenter study (AGEO RASANC) included 412 patients with metastatic CRC and paired tumor and plasma samples at diagnosis. CtDNA was evaluated by both NGS and a methylation-based PCR [62]. The accuracy of ctDNA assessment compared with tumor tissue reached 95%. Interestingly, the accuracy was as high as 97% among patients with CRLM (*n* = 293) by combining data from both NGS and methylation-based PCR. Patients without CRLM were more likely to have inconclusive ctDNA findings [62]. The same study has also reported a high accuracy of 97% for the detection of BRAF mutations among 405 patients with metastatic colorectal cancer. The respective percentage for the subset of patients with liver metastases reached 99% [63]. Similar results have been reported in the METABEAM study including RAS mutational data from 221 patients with stage IV CRC and available tissue and liquid biopsies [64]. Patients with liver-only metastases (*n* = 151) showed the highest concordance rate at 91% compared with those with peritoneum-only (88%) and lung-only metastases (64%) [64]. Another prospective trial that included 100 CRLM patients reported a 93% concordance rate on the detection of RAS/BRAF mutations between tumoral DNA and ctDNA [56]. The combination of the two approached increased the detection of mutations in 60% of the patients. A cost-effective approach would be to perform tissue biopsy in case of consecutive negative ctDNA in order to rule out false negative results [56]. In smaller studies the concordance rates of the detection of RAS mutations between conventional biopsies from primary site or liver metastases and liquid biopsies may be lower but remains above 75% [53,57,65]. Although the NGS analysis of ctDNA reflects the genomic alterations in liver metastases [65], tissue biopsies from primary sites, liver biopsies from metastatic sites and liquid biopsies may provide complimentary information about the genomic landscape of colorectal carcinoma in each patient [66].

CTCs have been also evaluated for the characterization of RAS mutational status. According to available studies, the main limitation of this approach is the small number of CTCs detected in the peripheral blood, which may lead to inconclusive results [67]. Interestingly, allele-specific blocker PCR has been suggested as a very sensitive method to detect KRAS and BRAF mutations even in samples with two CTCs available [67]. Similar to ctDNA, CTCs may better represent the molecular pattern of liver metastases compared with the primary tumor site [68]. Preliminary evidence suggests that ctDNA and CTCs may provide complimentary data on the presence of KRAS, BRAF and PIK3CA mutations, but it has to be determined in larger studies [69].

## 4. Challenges and Conclusions

Although the prospect of liquid biopsies is promising, their best use has not been fully determined [42]. In cases in which the number of CTCs is increasing after a round of chemotherapy or ctDNA becomes detectable after treatment (i.e., no response or progression of disease), this might create additional stress to CRLM patients when no alternative, effective therapeutic options exist. CtDNA and CTC assays need more standardization; currently, there is no universally accepted assay to be used in clinical practice. In addition, investigators have used different assays in research studies and no head-to-head comparison of commercially available assays has been performed to date [30]. Furthermore, there is evidence to suggest that cfDNA clearance is dependent on renal function [33], thus interpreting ctDNA analyses among certain patients with cancer and renal dysfunction warrants caution. Optimizing assay sensitivity and specificity by taking also into account the probability of false positive results due to clonal hematopoiesis of indeterminate potential are necessary in order to determine the clinical utility of liquid biopsies in practice [45].

In conclusion, liquid biopsies have opened new avenues for cancer diagnostics, including improved prognostication, evaluation of response to treatment, early detection of relapse and detection of MRD, as well as monitoring of tumor evolution in the context of cancer therapies (Figure 2). Early results of liquid biopsy testing for common mutations offer an attractive and cost-saving complementary approach to the current tissue-based DNA mutation analysis. In this context, repeat tissue biopsy could be reserved only for indeterminate or false negative results of liquid biopsies. Serial postoperative assessment of ctDNA could help identify recurrences earlier, especially in the setting of indeterminate imaging surveillance findings. Large scale prospective trials are needed to further characterize and establish the role of liquid biopsies in the treatment of patients with CRLM.

## Figures and Tables

**Figure 1 cancers-14-04237-f001:**
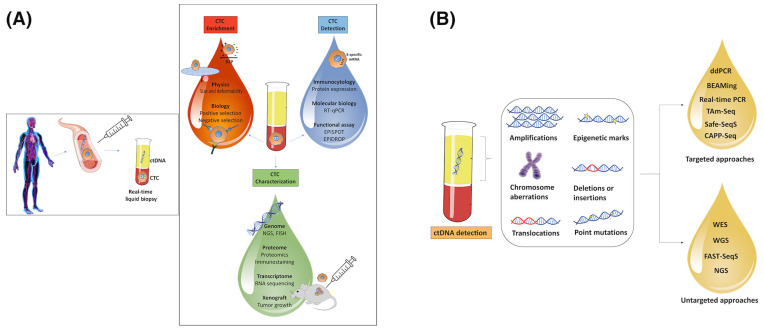
(**A**) Schematic representation of enrichment, detection and characterization of circulating tumor cells (CTCs). (**B**) ctDNA detection technologies. ctDNA analysis is based on the identification of tumors’ specific aberrations or epigenetic marks in cfDNA samples. Targeted approaches allow pre-specified cancer-associated mutations, whereas untargeted approaches facilitate detection of genomic aberrations without requiring prespecified information about the mutation pattern of the primary tumors. (Reprinted with permission from Heidrich et al. [30], Copyright 2020 International Journal of Cancer, open access article published under Creative Commons Attribution license (CC BY 4.0).

**Figure 2 cancers-14-04237-f002:**
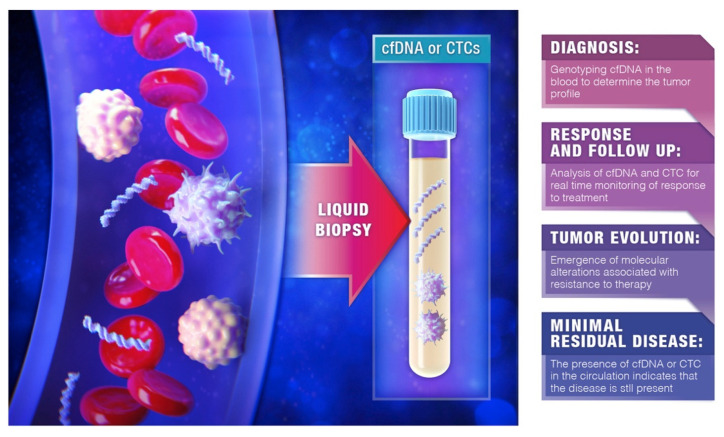
Clinical application of liquid biopsies (Reprinted with permission from Bardelli et al. [39]) Copyright 2022, Copyright Owner: Diamantis Tsilimigras).

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
