# Peer review of "Liquid Biopsies in Colorectal Liver Metastases: Towards the Era of Precision Oncologic Surgery"

_cancers, 2022, doi:10.3390/cancers14174237_

Round 1

Reviewer 1 Report

The authors have set up a review about the application of liquid biopsy in the management of colorectal cancer liver metastasis. They have nicely structured their work which is written in very good English language. 

The review needs some improvement though in mainly two respects:

1.     Not all recent literature has been cited (see below).

2.     At least once in the introduction all (so far) existing liquid biopsy should be mentioned and explained. Moreover, the focus of this review should be stated as it only covers CTCs and ctDNA, touching methylation once briefly. Other aspects are left aside without mentioning this.

Some detailed points:

Cited literature

Reviews missing:

2022: Raza et al. (PMID: 35292091 ) for novel LB markers, Zhou et al. (PMID: 35337361 ) for novel markers and similar content

2021: Gao et al. (PMID: 33584829) for similar review; Patelli et al. (PMID: 33738696) for CTCs vs. ctDNA; Galindo-Pumarino et al. (PMID: 33800796) for metCRC and LB and microenvironment

May be mentioned as well: Original article (2013): Groot Koerkamp et al. (PMID: 23456317) for CTCs in CRC liver mets.

2.1 “A high numbers of CTCs in patients with metastatic CRC has also been associated with adverse patient and disease characteristics.16” What does “adverse patient… characteristics” mean?

3.3 The results of the methylation marker study are not clearly described. 

3.4 

“CTCs have been also evaluated for the characterization of RAS mutational status. According to available studies, the main limitation of this approach is the small number of CTCs detected in the peripheral blood, which may lead to inconclusive results.”

References are needed for these statements.

4. “Knowing that the number of CTCs is increasing after a round of chemotherapy…”. Where does this come from? Which references account for this? This phenomenon standing for treatment failure/inefficiency needs a short explanation.

Author Response

Reviewer #1:

The authors have set up a review about the application of liquid biopsy in the management of colorectal cancer liver metastasis. They have nicely structured their work which is written in very good English language.

We thank the Reviewer for their valuable review and for suggesting ways to further improve our work.

The review needs some improvement though in mainly two respects:

  1. Not all recent literature has been cited (see below).

We thank the Reviewer for this comment. We have added the pertinent literature in our revised manuscript, as requested below.

  1. At least once in the introduction all (so far) existing liquid biopsy should be mentioned and explained. Moreover, the focus of this review should be stated as it only covers CTCs and ctDNA, touching methylation once briefly. Other aspects are left aside without mentioning this.

We thank the Reviewer for this comment. The definition of liquid biopsy has been better explained in the Introduction section. In addition, the authors have added pertinent sentence to further report and explain all different types of liquid biopsies in the Introduction as requested by the Reviewer. Furthermore, the focus of this review (CTCs and ctDNA) has been further highlighted in the Introduction section of our revised manuscript, as requested.

Some detailed points:

Cited literature

Reviews missing:

2022: Raza et al. (PMID: 35292091) for novel LB markers, Zhou et al. (PMID: 35337361 ) for novel markers and similar content

2021: Gao et al. (PMID: 33584829) for similar review; Patelli et al. (PMID: 33738696) for CTCs vs. ctDNA; Galindo-Pumarino et al. (PMID: 33800796) for metCRC and LB and microenvironment

The above-mentioned literature has been cited in the revised version of our manuscript, as requested

May be mentioned as well: Original article (2013): Groot Koerkamp et al. (PMID: 23456317) for CTCs in CRC liver mets.

We thank the Reviewer for this comment. The above-mentioned meta-analysis has also been added in the revised version of our manuscript, as requested.

2.1 “A high numbers of CTCs in patients with metastatic CRC has also been associated with adverse patient and disease characteristics.16” What does “adverse patient… characteristics” mean?

We thank the Reviewer for this comment. A high number of baseline CTCs in patients with metastatic CRC has been associated with adverse patient and disease characteristics, including worse performance status, stage IV at diagnosis, at least 3 metastatic sites, and elevated CEA levels based on the cited literature. This has been further explained in the relevant sentence as appropriate.

3.3 The results of the methylation marker study are not clearly described.

We thank the Reviewer for this comment. The authors have now added a new paragraph that thoroughly describes the results of the methylation marker study, as requested.

3.4 “CTCs have been also evaluated for the characterization of RAS mutational status. According to available studies, the main limitation of this approach is the small number of CTCs detected in the peripheral blood, which may lead to inconclusive results.”

References are needed for these statements.

We thank the Reviewer for this comment. Pertinent citation has been added to support the above statement.

  1. “Knowing that the number of CTCs is increasing after a round of chemotherapy…”. Where does this come from? Which references account for this? This phenomenon standing for treatment failure/inefficiency needs a short explanation.

We thank the Reviewer for this comment. We apologize for the confusion. The author meant to state that in case the number of CTCs increases after a round of chemotherapy (not that chemotherapy necessarily increases CTCs), likely due to no response to treatment, that could create additional stress to patients when no alternative, effective therapeutic options exist. The authors have revised the above-mentioned statement as appropriate.

Reviewer 2 Report

This is a brief, but useful review of the current work on liquid biopsies in CRLM and will fit well into the Special Issue on CRC metastasis.  The review of CTCs and ctDNA in liquid biopsies is necessarily brief but accurate as far as it goes. In addition to enrichment and depletion methods for CTCs the authors could also mention the distinctly different non-enrichment methodology used by Epic Sciences Inc. for the CLIA-CAP approved and commercially available ARv7 assay used for predicting treatment switching in metastatic prostate cancer. Also in Figure 1a, the text within the artsy 'droplets' is very hard to read. I suggest lightening the background colors so that the print stands out, and making the print a bit larger.

The treatment of current work on mutant identification in CRLM is reported accurately and presented in proper context. 

An adjustment of Figure 1a would be useful for the reader, otherwise the paper can be accepted in the current form.

Author Response

Reviewer #2:

This is a brief, but useful review of the current work on liquid biopsies in CRLM and will fit well into the Special Issue on CRC metastasis.  The review of CTCs and ctDNA in liquid biopsies is necessarily brief but accurate as far as it goes. In addition to enrichment and depletion methods for CTCs the authors could also mention the distinctly different non-enrichment methodology used by Epic Sciences Inc. for the CLIA-CAP approved and commercially available ARv7 assay used for predicting treatment switching in metastatic prostate cancer. Also in Figure 1a, the text within the artsy 'droplets' is very hard to read. I suggest lightening the background colors so that the print stands out, and making the print a bit larger.

We thank the Reviewer for their valuable review and for suggesting ways to further improve our work. The authors have mentioned the Arv7 assay and the different non-enrichment methodology used for predicting treatment in metastatic prostate cancer as requested. In addition,  The authors have provided the Journal with a magnified form of the figure (adopted with permission from another journal) and have lightened the background color of the image as requested. The authors have asked the Journal to provide the figure in a magnified form in the final version of our paper.

The treatment of current work on mutant identification in CRLM is reported accurately and presented in proper context. An adjustment of Figure 1a would be useful for the reader, otherwise the paper can be accepted in the current form.

We thank the Reviewer for this comment. The authors have provided the Journal with a edited form of the figure (lightened background) magnified form of the figure (adopted with permission from another journal) and have asked the Journal to provide the figure in a magnified form in the final version of our paper.

Thank you for considering our revised manuscript.

Sincerely,

Timothy M. Pawlik, MD, MPH, MTS, PhD, FACS, FRACS (Hon.)

Professor and Chair, Department of Surgery

The Urban Meyer III and Shelley Meyer Chair for Cancer Research

Professor of Surgery, Oncology, and Health Services Management and Policy

Surgeon in Chief, The Ohio State University Wexner Medical Center

The Ohio State University, Wexner Medical Center

Round 2

Reviewer 1 Report

Dear Authors,

This revision of your manuscript is very good. 

Kind regards!